



# High resolution wind speed measurements with quadcopter UAS: calibration and verification in a wind tunnel with active grid

Johannes Kistner[1], Lars Neuhaus[2,3], and Norman Wildmann[1]

[1]Deutsches Zentrum für Luft- und Raumfahrt, Institut für Physik der Atmosphäre, Oberpfaffenhofen, Germany
[2]Carl von Ossietzky Universität Oldenburg, School of Mathematics and Science, Institute of Physics
[3]ForWind - Center for Wind Energy Research, Küpkersweg 70, 26129 Oldenburg, Germany

**Correspondence:** Johannes Kistner (johannes.kistner@dlr.de)

**Abstract.** As a contribution to closing observational gaps in the atmospheric boundary layer (ABL), the SWUF-3D fleet of unmanned aerial systems (UAS) is utilized for in situ measurements of turbulence. To date, the algorithm for wind measurement has only been calibrated in the free field. We therefore present in this work the calibration and verification under laboratory conditions. The UAS measurements are performed in a wind tunnel with active grid and constant temperature anemometers (CTAs) as a reference. Calibration is performed in $x$- and $y$-coordinate directions of the UAS body frame in wind speeds of $2\ldots18$ m s$^{-1}$. For systematic verification of the measurement capabilities and identification of limitations, different measurement scenarios like gusts, velocity steps and turbulence are generated with the active grid. Furthermore, the measurement accuracy under different angles of sideslip (AoS) and wind speeds is investigated and it is examined whether the calibration coefficients can be ported to other UAS of the fleet. Our analyses show that the uncertainty depends on the wind speed magnitude and increases with higher wind speeds, resulting in an overall root-mean-square error (RMSE) of less than $0.2$ m s$^{-1}$. Applying the calibration coefficients from one UAS to others within the fleet results in comparable accuracies. Flights in different gusts yield an RMSE of up to $0.6$ m s$^{-1}$. The maximal RMSE occurs in the most extreme velocity steps (i.e. a lower speed of $5$ m s$^{-1}$ and an amplitude of $10$ m s$^{-1}$) and exceeds $1.3$ m s$^{-1}$. For variances below approx. $0.5$ m$^2$ s$^{-2}$ and $0.3$ m$^2$ s$^{-2}$, the maximal resolvable frequencies of the turbulence are about $2$ Hz and $1$ Hz respectively. We report successful calibration but with susceptibility to high AoS in high wind speeds, no necessity of wind tunnel calibration for individual UAS, and the need for further research regarding turbulence analysis.

## 1 Introduction

Unmanned aerial systems (UAS) are steadily gaining popularity for measuring the wind in the atmospheric boundary layer (ABL). There are several approaches to measuring the wind vector (i. e. wind speed and direction), which can be grouped into two main categories: First, the direct wind measurement, in which the UAS used as a sensor platform carries an anemometer (e.g. ultrasonic anemometer, multi hole probe, lidar, hot wire anemometer) as a payload. These UAS can be fixed-wing aircraft (Wildmann et al., 2015; Platis et al., 2018; Rautenberg et al., 2019; zum Berge et al., 2021) or multicopters (Palomaki et al., 2017; Shimura et al., 2018; Nolan et al., 2018; Molter and Cheng, 2020). The second approach is the indirect wind measurement, in which the avionic data of the UAS (e.g. Euler angles, accelerations, motor thrust, etc.) are used to derive the wind





that a multicopter UAS is exposed to during hover (Palomaki et al., 2017; Simma et al., 2020; Shelekhov et al., 2022) or in steady flight (Brosy et al., 2017; Segales et al., 2020; Hattenberger et al., 2022; González-Rocha et al., 2023). For this measurement method, knowledge of particular aerodynamic or flight dynamic characteristics of the respective UAS is necessary, which is usually not given a priori. This is why a calibration of the wind measurement with the respective UAS model becomes necessary.

The wind estimation with the SWUF-3D (Simultaneous Wind measurement with Unmanned Flight Systems in 3D) fleet (Wetz et al., 2021; Wildmann and Wetz, 2022) for determination of turbulence in the ABL (Wetz and Wildmann, 2023; Wetz et al., 2023) is also based on the latter technique. The so far used calibration of the fleet and its validation have been performed in free field measurements up to this point (Wetz and Wildmann, 2022), which always have inherent uncertainties. This motivates the first part of this work: the calibration of the coefficients of the extended Rayleigh drag equations of the wind measurement

algorithm from Wetz and Wildmann (2022) under laboratory conditions. The calibration is carried out for mean wind speeds along the main axes ($x$- and $y$-direction) of the fixed-body coordinate system of a single UAS.

    Calibration for the fleet based on only one UAS is based on the idea that by porting the determined calibration coefficients to other UAS of the same type, but with potential production variations (e.g. autopilot orientation), these are also capable of accurately measuring the wind. Furthermore, this approach is subject to the hypothesis that calibration along the longitudinal

and lateral axes of the UAS will allow reliable wind measurement at angles of sideslip (AoS) off those axes, i.e., applicability in the case of divergent inflow conditions. More generally, the overall approach is based on the hypothesis that it is possible to reliably measure highly dynamic flows with the UAS via calibration in averaged wind speeds. For this purpose, we perform verification in reproducible flow scenarios designed to cover real-world application, along with a systematic determination of the limitations of the turbulence measurement.

Wind tunnel measurements for comparable purposes have already been carried out before: Hattenberger et al. (2022) performed their measurements not in a wind tunnel but indoor with a wind generator. Wang et al. (2018) also verified in an indoor experiment but at very low, constant velocities ($< 2$ m s$^{-1}$). Thielicke et al. (2021) performed wind tunnel calibration and testing of their system as well, which however features an ultrasonic anemometer attached to a multicopter, and thus counts to the direct wind measurement technique. González-Rocha et al. (2019) conducted wind tunnel tests as part of their review of

various indirect measurement techniques, but only for the rotor and not the entire UAS. Marino et al. (2015) carried out wind tunnel calibrations for their measurement method using the power consumption of the rotors, but they conclude its practicality to be limited. The closest related work is the research of Neumann and Bartholmai (2015). They determined the drag coefficient of their UAS in the wind tunnel at sideslip angles of $0°$, $45°$ and $90°$ to the main wind direction with wind speeds up to $8$ m s$^{-1}$, in order to be able to use it to obtain the wind speed via the Rayleigh drag equations and the system described by Neumann

et al. (2012).

    Going beyond that, the work presented in this paper aims at the determination of the additional calibration coefficients of the extended Rayleigh drag equation of the wind measurement algorithm from Wetz and Wildmann (2022) for a significantly extended velocity envelope. This and the systematic investigation in the wind tunnel of applicability in more complex flow conditions, of applicability in the case of several divergent inflow conditions off the main axes, and of applicability to the



entire fleet distinguish our work from the state of research. We highlight that this is the first time of wind measurements with a multicopter UAS in the reproducible turbulent inflow fields of a wind tunnel with active grid. For these studies, the underlying hypotheses mentioned above lead to the following research questions:

1. What are the accuracies at different angles of sideslip?

2. What are the measurement accuracies for different UAS using the same calibration coefficients?

3. What are the accuracies in different more complex flow conditions in terms of wind speed, response time behavior and resolution?

Thus, this study aims to gain enhanced data for calibration along with the systematic detection of measurement limitations or potential situations in which the wind measurement method is subject to higher uncertainties. It is not intended to replace validation in the open field or to give conclusions about measurement capabilities when measuring with the entire fleet. In the 70 next chapter a description of our UAS and the wind measurement system as well as the wind tunnel including the reference sensors is given. In chapter 3 the procedures for calibration and verification are explained. The results are presented in the fourth chapter and discussed in chapters 5 and 6.

## 2  System description

Our experimental setup consists of the UAS, with which we measure against reference sensors in a wind tunnel with an active 75 grid. These systems are described in more detail below. The full setup used is shown in Fig. 1.

### 2.1  UAS and wind measurement

The type of UAS in this project is a quadcopter. It is based on the Holybro QAV250 airframe and the Pixhawk autopilot, running a custom version of the PX4 firmware (Table 1). The copter measures $0.25$ m from axis to axis and weighs $0.645$ kg including battery. A flight duration of up to $15$ min can be achieved, depending on wind conditions. For a more detailed description of 80 the hardware, refer to Wetz et al. (2021) and Wetz and Wildmann (2022).




**Table 1.** System description of the used UAS

| Parameter | Description |
| --- | --- |
| copter type | quadrotor |
| airframe | Holybro QAV250 |
| autopilot | Pixhawk 4 Mini |
| firmware version | v13.3.0 dev |
| weight (including battery) | 0.645 kg |
| dimension (axis to axis) | 0.25 m |
| max. flight time | 15 min |
| max. tested flight speed | 18 m s$^{-1}$ |

Wind measurements are made by the UAS hovering in one place and always turning its nose automatically into the wind by applying PX4's weather vane mode. During this operation, the attitude of the UAS is tracked by measuring and logging the avionic data with a temporal resolution of 1 to 250 Hz, depending on the particular sensor. By applying the wind measurement algorithm, which is using an adaption of the Rayleigh drag equation (Eq. 1, see Wetz et al. (2021); Wetz and Wildmann (2022) for a more detailed description), to the measured attitude and its first and second derivatives, the wind acting on the UAS during hover can be determined. More precisely, the wind vector

$$\boldsymbol{V} = \begin{pmatrix} u \\ v \end{pmatrix}_g = \mathbf{R} \begin{pmatrix} c_x \cdot F_{\mathrm{w,x}}^{b_x} \\ c_y \cdot F_{\mathrm{w,y}}^{b_y} \end{pmatrix}_b - \begin{pmatrix} \dot{x}_{\mathrm{gps}} \\ \dot{y}_{\mathrm{gps}} \end{pmatrix}_g , \tag{1}$$

consisting of the longitudinal wind speed component $u$ and the lateral component $v$ in the geodetic coordinate system, is calculated from the wind forces $F_{\mathrm{w,i}}$ determined from the avionic sensors and the calibration coefficients $c_i$ and $b_i$. Since the forces and coefficients are given in the body-fixed coordinate system, they have to be converted into the geodetic coordinate system using the rotation matrix $\mathbf{R}$ based on the Euler angles pitch $\theta$, roll $\phi$ and yaw $\psi$ (Eq. 2). Motions of the UAS that deviate from a stationary hover are detected by the GNSS ($\dot{x}_{\mathrm{gps}}$, $\dot{y}_{\mathrm{gps}}$) and corrected in the calculation. This means that wind speed and wind direction can be measured without an additional wind sensor.

$$\mathbf{R} = \begin{bmatrix} \cos\theta\cos\psi & \cos\psi\sin\theta\sin\phi - \cos\phi\sin\psi & \cos\psi\sin\theta\cos\phi + \sin\phi\sin\psi \\ \cos\theta\sin\phi & \cos\phi\cos\psi + \sin\theta\sin\phi\sin\psi & -\sin\phi\cos\psi + \sin\theta\cos\phi\sin\psi \\ -\sin\theta & \cos\theta\sin\phi & \cos\theta\cos\phi \end{bmatrix} \tag{2}$$

The sensors used include accelerometers, gyroscopes, magnetometers, barometers and GNSS receivers in the free field. Since sufficient GNSS signal reception is not available in the wind tunnel, an optical positioning system is used instead. This consists of the Hex HereFlow optical flow sensor for holding the position in longitudinal and lateral earth coordinate directions, and the Benewake TF-Mini rangefinder for holding the altitude over ground. These sensors are very small and lightweight, so their effects on drag and weight are negligible. The rangefinder performs sufficiently well. However, the used optical flow





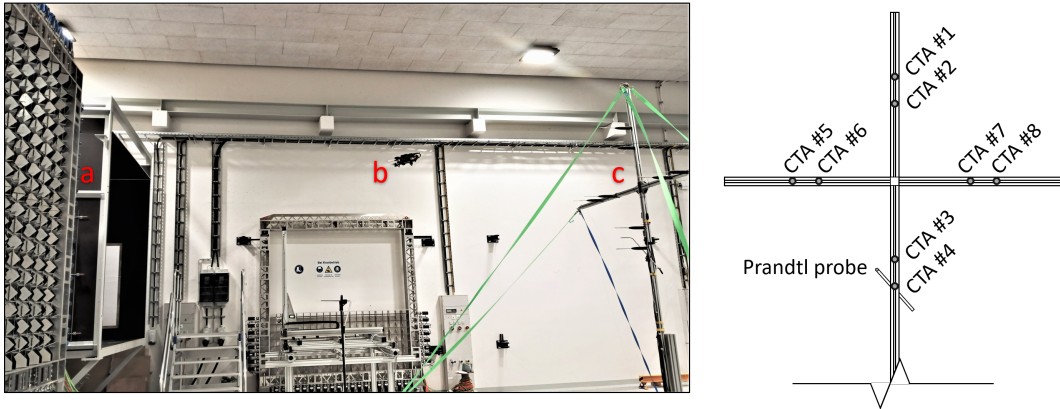

**Figure 1.** left: measurement setup in the wind tunnel with **a.** active grid, **b**. UAS, and **c**. reference sensor cross; right: schematic front view of the reference sensor cross with positions of the CTAs and Prandtl probe.

sensor type has an inherent imprecision, which causes the UAS to drift away while hovering. In calm air, the positional drift is about 1 m per minute. As wind speeds increase, the intensity and direction of the drift change without a discernible systematic, which required constant adjustment counteracting the drift during the test flights. These adjustments were executed by the remote pilot through manual trim.

## 2.2 Wind tunnel

The measurements are performed in the turbulent wind tunnel in Oldenburg (Kröger et al., 2018). The wind tunnel has a test section that measures $3 \times 3$ m$^2$ with a length of 30 m. In the open configuration, wind speeds up to 32 m s$^{-1}$ can be achieved by 4 fans with 110 kW each. The centerpiece of the turbulent wind tunnel is the active grid, which is attached to the nozzle. The active grid covers the entire area of $3 \times 3$ m$^2$ and consist of 80 individual controlled shafts with a mesh width of 0.143 m. The active grid allows to change its blockage dynamically and locally from 21 % to 92 %. To tailor turbulent flows and generate specific flow structures, the blockage induced flow design approach is used (Neuhaus et al., 2021). Based on a transfer function between shaft angle and wind speed, specific shaft angle time series can be determined for individual shaft motion groups. In this way, the approach allows for very strong velocity variations on short time scales over a large cross-sectional region. The shaft motion groups chosen in this work produce approximately homogeneous flow characteristics over an area of $2 \times 2$ m$^2$. Within this area and the main measurement zone of the UAS with the active grid being opened completely and inoperative, we determined a standard deviation of 0.06 m s$^{-1}$ and 0.14 m s$^{-1}$ for 1 minute averages of wind speeds at 5 m s$^{-1}$ and 10 m s$^{-1}$ respectively, whereby the deviation in lateral direction is the higher towards the grid. To excite the largest scales, an additional dynamic excitation is done by the wind tunnel fans (Neuhaus et al., 2020).





## 2.3 Reference sensors

As reference sensors, constant-temperature anemometers (CTAs) and a Prandtl probe are used. The CTA setup consists of 1D
hot-wire sensors on multichannel CTAs by Dantec Dynamics mounted on a cross of system profiles (Fig. 1), which is located
at a distance of 5 m from the outlet of the wind tunnel and centered in the flow cross section. The crossbeam is at the flight
level of the UAS, which is why CTAs no. 5, 6, 7, and 8 may experience temporal interference from the wake of the UAS.
Furthermore, CTAs no. 3 and 4 may experience perturbations due to the downwash of the UAS. Test runs with no UAS show
that all CTAs measure the equivalent wind speeds with sufficient accuracy: the standard deviation of the measured wind speed
of the individual CTAs is less than $0.05$ m s$^{-1}$ in the calibrations, and $0.16$ m s$^{-1}$ when turbulence is actively excited by the
active grid. Therefore, it is acceptable to use one of the undisturbed CTAs as a reference, although they do not measure at the
same height in the flow as the UAS does. The sensor selected as reference is CTA no. 2. The CTAs measure wind speeds at
frequencies from 100 Hz to 6 kHz in the experiments presented in this work. For a better comparability the measurement data
in all measurements, except those dedicated to the analysis of turbulence, are sampled down to the frequency to which the
measurement data of the UAS are filtered down to, i.e. 10 Hz.

Each morning and evening, the CTAs are calibrated in the flow against the Prandtl probe which is also mounted on the cross
at the height between CTAs no. 3 and 4. For the calibration of the CTAs, 16 steps of logarithmically increasing wind speeds
between 2 and 20 m s$^{-1}$ are set for 30 s each. Based on the measured ambient conditions, the wind speeds at these steps are
determined using the Prandtl probe. The simultaneously measured voltages of the CTAs are then translated into velocities.
Careful quality checks were carried out for the CTA measurement data and corrupted data was sorted out.

## 3 Methods

For an improved calibration of the wind measurement algorithm, flights are performed with the UAS in the wind tunnel and
based on the measured data a calibration according to Wetz and Wildmann (2022) is carried out. In order to determine the
accuracy of the wind measurement, verification flights are performed with the UAS in different flow scenarios.

During the measurement flights, the UAS hovers at all times at an altitude of 3 m (i.e., the vertical center of the flow) and at
half distance in longitudinal direction between the outlet of the wind tunnel and the CTAs (i. e. 2.5 m), centered in the lateral
direction of the main flow.

### 3.1 Calibration

The procedure for calibrating the wind measurement algorithm to find the calibration coefficients $c_x, c_y, b_x$ and $b_y$ for Eq. 1
involves a step-wise linear increase in wind speed of 2 m s$^{-1}$ starting from 2 m s$^{-1}$ to the maximum speed $V_{\max}$. Each step
is held for the time $T_s$ (see Table A1). For each interval, we take the average of the wind speed measured by the CTA and
the accelerations of the UAS measured by the autopilot's avionics. For receiving the transfer function between the respective
acceleration and wind speed averages, an optimization algorithm is used to find the calibration coefficients in the adapted





Rayleigh drag equation (Eq. 1) which best fit the accelerations to the reference wind speeds. This principle is comparable to the procedure from ISO 17713-1:2007 and Wetz and Wildmann (2022), in which a transfer function is set up between the averaged values of a reference sensor system and the anemometer to be calibrated.

It is fundamental to calibrate using the accelerations in the longitudinal direction of flight due to the weather vane mode. In order to compensate minor errors between the heading of the UAS and the wind direction, the calibration is also performed in positive and negative lateral direction of the UAS. For this purpose, the UAS is calibrated with $+90°$ and $-90°$ yaw angle to the flow direction, while the weather vane mode remains deactivated. The calibration is performed on the measurement data of 6 flights with the UAS heading parallel and 4 flights orthogonal to the main flow direction.

### 3.2 Verification

The calibration is carried out on an individual UAS of the fleet, i.e. UAS no. 6. It is to be verified that this calibration can be ported to other UAS of the fleet without significant loss of accuracy. Furthermore, the calibration is performed in stationary wind speeds with the wind direction following the main coordinate directions of the UAS body frame. For verification of the applicability of the calibration outside the calibrated range, measurements with different AoS and in more dynamic flow (i.e. gusts, velocity steps, turbulence) are performed. The results are analyzed with respect to the error of the wind measurement compared to the reference, the response time behavior and the achieved temporal resolution.

#### 3.2.1 Portability

The SWUF-3D fleet currently consists of 35 UAS with identical design. This suggests that each UAS of the fleet should show the same flight physics and behavior. In this case, the calibration coefficients of one UAS are portable to the rest of the fleet, without a necessity to individually calibrate each UAS. For verification of this portability, we apply the wind measurement algorithm using the calibration coefficients of UAS no. 6 (cf. subchapter *Calibration*) to the measurement data of other UAS in the fleet and determine the accuracy of their wind measurement, using the root-mean-square error (RMSE) $\epsilon$:

$$\epsilon_V = \sqrt{\frac{1}{n}\Sigma_{i=1}^{n}(V_{i,\mathrm{UAS}} - V_{i,\mathrm{CTA}})^2} \tag{3}$$

We execute individual measurements for UAS no. 31, 33, 34 and 35 with the calibration pattern and compare their measurement accuracies with the accuracy of UAS no. 6. If the portability of the coefficients is given, only the individual accelerometer offset for each UAS has to be determined, which is due to minor installation-related differences of the autopilot's orientation within the UAS. For this purpose, no wind tunnel is required (cf. Wetz and Wildmann (2022)).

#### 3.2.2 Angles of sideslip

The weather vane mode adjusts the yaw angle of the UAS to the mean wind direction. Rapid changes due to small-scale turbulence are not accounted for. It is thus important to investigate the impact of AoS (a detailed list of AoS settings and the number of runs $r$ can be found in Table A1) on the horizontal wind measurement. In this context, smaller angles and errors in



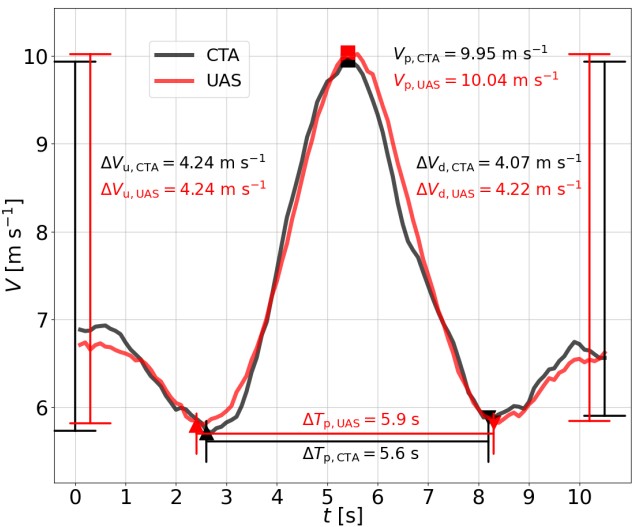

**Figure 2.** Exemplary wind speed time series comparison of a gust measured by the UAS and the CTA. The black line represents the time series measured with the CTA, the red line represents the time series measured with the UAS.

the lower wind speeds are of particular relevance. To maintain the AoS throughout the test, we disable the weather vane mode.
For wind measurements at various AoS, we use the wind tunnel program of the calibration flights.

### 3.2.3 Gusts

Calibration is carried out with steps of constant wind speed over an averaging period of 30 s resp. 60 s (see Sect. 3.1). In order to verify the measurement behavior in more dynamic and more realistic flow conditions, measurements are carried out in gusts (Fig. 2). The definition of the gusts is based on IEC 61400-1:2019, using the gust profile

$$V = V_0 - 0.37 \, V_g \, \sin\left(\frac{3\pi t}{2T_{\text{peak}}}\right)\left(1 - \cos\left(\frac{2\pi t}{2T_{\text{peak}}}\right)\right) \tag{4}$$

with the initial wind speed $V_0$ , the gust speed $V_g$ and the gust duration $T_{\text{peak}}$.

From the set of resulting gust profiles, the most and least distinct gusts are selected for the different wind speed ranges. This yields the gusts listed in Table A2, of which 10 repetitions were measured in each flight.

The duration of a single gust is 10.5 s. We analyse the wind measurement for the initial and maximum wind speeds $V_0$ and
$V_p$ (i.e., $V_0 + V_g$) via Eq. 3, and the timing accuracy by comparing the time delta $\Delta T$ between the downward peaks registered by the UAS and the CTA:

$$\epsilon_{\Delta \mathrm{T}} = \sqrt{\frac{1}{n} \Sigma_{i=1}^{n} (\Delta T_{\mathrm{i,UAS}} - \Delta T_{\mathrm{i,CTA}})^2} \tag{5}$$



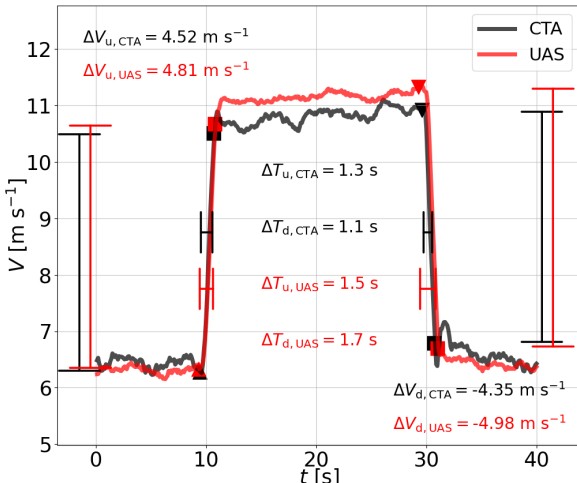

**Figure 3.** Exemplary wind speed time series comparison of velocity steps measured by the UAS and CTA. The black line represents the time series measured with the CTA, the red line represents the time series measured with the UAS.

### 3.2.4 Velocity steps

Similar to the analysis of the measurement in gusts, tests are performed in velocity steps (Fig. 3) to verify the response time behavior and the accuracy of the wind measurement in abrupt changes of the flow condition. For this study the flow generation configurations from Neuhaus et al. (2021) are used. Therefore, the following setups for the initial velocity $V_0$ and the step velocity $V_{\mathrm{du}}$, which lie within the flight envelope of the UAS, are used (Table A3). For the step amplitudes 7 m s$^{-1}$ and 10 m s$^{-1}$, 10 repetitions were performed, and for the step amplitude of 5 m s$^{-1}$, a total of 20 repetitions were performed. Each repetition as shown in Fig. 3 lasts 40 s. They consist of one upward and one downward velocity step with equal duration of the intervals of constant velocity between the steps. In order to measure the time response, the start time of the step and the final time when the target wind speed is reached need to be detected from the time series. The starting points $T_{\mathrm{u},0}$ and $T_{\mathrm{d},0}$ of the upward and downward steps are defined from the most extreme gradients in the time series:

$$T_{\mathrm{u},0} = t(\max(\dot{V})) \tag{6}$$

$$T_{\mathrm{d},0} = t(\min(\dot{V})) \tag{7}$$

The final points in time of the steps $T_{\mathrm{u},1}$ and $T_{\mathrm{d},1}$ are defined when 90 % of the delta between the measured higher and lower stationary velocities $V_h$ and $V_l$ are reached:

$$T_{\mathrm{u},1} = t(0.9(V_h - V_l)) \tag{8}$$

$$T_{\mathrm{d},1} = t(1.1(V_h - V_l)) \tag{9}$$





The measured upward and downward velocity steps $\Delta V_u$ and $\Delta V_d$ as well as the step times $\Delta T_u$ and $\Delta T_d$ are defined through these points in the time series:

$$\Delta V_h = V(T_{u,1}) - V(T_{u,0}) \tag{10}$$

$$\Delta V_d = V(T_{d,1}) - V(T_{d,0}) \tag{11}$$

$$\Delta T_u = T_{u,1} - T_{u,0} \tag{12}$$

$$\Delta T_d = T_{d,1} - T_{d,0} \tag{13}$$

We analyse the timing accuracy by comparing these time deltas using Eq. 5. The accuracy of measurements by the UAS and the CTA of $V_l$ and $V_h$ are analysed via Eq. 3 and of the upward and downward velocity steps $\Delta V_u$ and $\Delta V_d$ are analysed with Eq. 14:

$$\epsilon_{\Delta V} = \sqrt{\frac{1}{n}\Sigma_{i=1}^{n}(\Delta V_{i,\mathrm{UAS}} - \Delta V_{i,\mathrm{CTA}})^2} \tag{14}$$

### 3.2.5 Turbulence

To verify the capability of the UAS to resolve turbulence, we perform experiments with different turbulence intensities and mean wind speeds. For this, we use two programs for the active grid's flap motion control, one for generating higher turbulence intensity and one for lower turbulence intensity. We run a constant wind speed in the wind tunnel and execute the programs. In addition, the wind tunnel controller can use the speed of the wind tunnel's fans. This enables achieving higher length scales and intensities of turbulence. The setups of preset wind speed $V_0$ and turbulence intensity $I$ are listed in Table A4. The setting $V(t)$ means that no constant background wind speed was set, but the wind tunnel fans were included in the turbulence generation. This turbulence generated with the active grid is reproducible from a statistical point of view and is therefore referred to as statistical turbulence.

Each measurement run for statistical turbulence has a duration of $600$ s. By examining the power spectral density (PSD) of the turbulence we investigate the maximum resolvable frequency $f$ as well as the determination of the turbulence intensity $I$ (Eq. 16) with the standard deviation $\sigma$ (Eq. 15) of all values $V_i$ of the wind speed time series. The RMSE in determination of the turbulence intensity $\epsilon_I$ is calculated and analyzed in dependence of the variance $\sigma^2$ (Eq. 17).

$$\sigma = \sqrt{\frac{1}{n}\Sigma_{i=1}^{n}(V_i - \overline{V})^2} \tag{15}$$

$$I = \frac{\sigma}{\overline{V}} \tag{16}$$

$$\epsilon_I = \sqrt{\frac{1}{n}\Sigma_{i=1}^{n}(I_{i,\mathrm{UAS}} - I_{i,\mathrm{CTA}})^2} \tag{17}$$



**Table 2.** Calibration Coefficients for different speed ranges

| speed range [m s$^{-1}$] | $b_x$ [−] | $c_x$ [−] |
|---|---|---|
| $(-18, 4]$ | 0.784 | 6.338 |
| $(4, 12)$ | 0.882 | 7.604 |
| $[12, 18)$ | 0.635 | 8.600 |

| speed range [m s$^{-1}$] | $b_y$ [−] | $c_y$ [−] |
|---|---|---|
| $(-18, 18)$ | 0.860 | 5.600 |

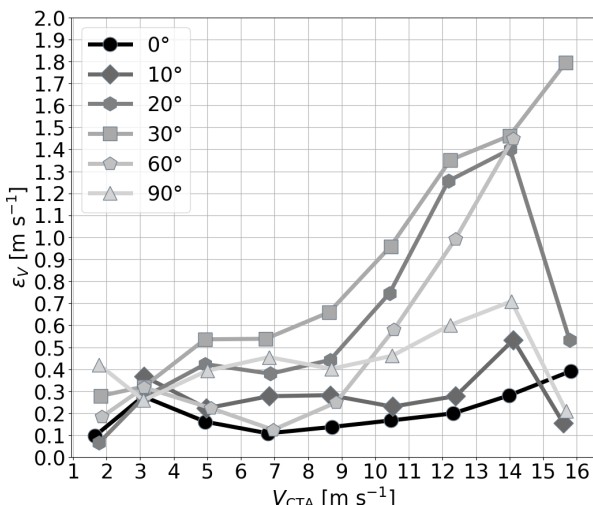

**Figure 4.** RMSE of UAS-based speed measurement $\epsilon_V$ over reference wind speed $V_{\mathrm{CTA}}$. The different lines represent various AoS during the measurements.

## 4 Results

**Calibration**. We have found that higher accuracy can be attained, particularly at high and low speeds, if we use dedicated calibration coefficients for different speed ranges in the longitudinal direction, as listed in Table 2.

**Angles of sideslip**. By applying these coefficients, we obtain the RMSE for the wind speed measurement $\epsilon_V$ for each speed step and each AoS (see Fig. 4). At 0° AoS, $\epsilon_V$ is below $0.4$ m s$^{-1}$ and below $0.2$ m s$^{-1}$ in the wind speed range $5\ldots12$ m s$^{-1}$. For all measured wind speeds, the overall $\epsilon_V$ yields $0.19$ m s$^{-1}$. At an AoS of 90° and wind speeds up to 11 m s$^{-1}$, $\epsilon_V$ is below $0.5$ m s$^{-1}$ and reaches a maximum of about $0.7$ m s$^{-1}$ at a wind speed of 14 m s$^{-1}$. For the other angles, especially 20°, 30° and 60°, the error is significantly higher and in particular above wind speeds of 9 m s$^{-1}$.



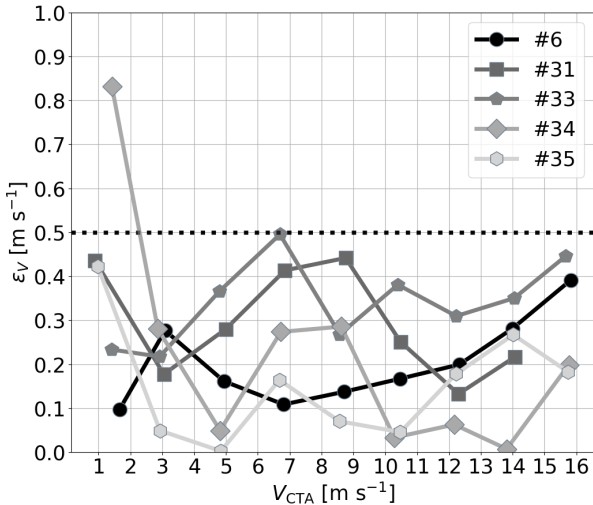

**Figure 5.** RMSE of UAS-based speed measurement $\epsilon_V$ over reference wind speed $V_{\mathrm{CTA}}$. The different lines represent different UAS used for measurement.

**Portability**. Applying the calibration coefficients to other UAS in the fleet, we again determine $\epsilon_V$ for each wind speed step (see Fig. 5). Throughout the fleet, $\epsilon_V$ is below $0.5$ m s$^{-1}$ for all velocity ranges, except for one data point: at a wind speed of $1.4$ m s$^{-1}$ UAS no. 34 shows an error of over $0.8$ m s$^{-1}$ compared to the CTA. Why UAS no. 34 experienced this acceleration despite the low wind speed can not be reproduced at this point and is likely due to a random error in the experimental setup.

**Gusts**. For the gust measurements, we analyse both, the accuracy of wind speed measurement and the response time behavior. The RMSE for the response time behavior $\epsilon_{\Delta T}$ is $4.5 \cdot 10^{-1}$ s with a reference sensor standard deviation of $3.6 \cdot 10^{-1}$ s . The RMSE of the wind speed measurement over the different speed ranges are shown in Fig. 6. For all $V_0$ and for $V_p$ up to $13.5$ m s$^{-1}$, $\epsilon_V$ is $\leq 0.5$ m s$^{-1}$.

**Velocity steps**. For the velocity steps, just like for the gusts, we investigate the accuracy of wind speed measurement and response time. Here $\epsilon_{\Delta T}$ for $\Delta T_u$ and $\Delta T_d$ is $7.1 \cdot 10^{-1}$ s with a reference sensor standard deviation of $6.5 \cdot 10^{-1}$ s. The RMSEs of the wind speed measurements are shown in Fig. 7. The $\epsilon_V$ for $V_l$ is about $0.8$ m s$^{-1}$ and for $V_h$ it is $\leq 0.3$ m s$^{-1}$. A significant increase in the error can be observed for $\Delta V_u$ and $\Delta V_d$ with $\epsilon_{\Delta V}$ of almost $1.0$ m s$^{-1}$ and over $1.3$ m s$^{-1}$, respectively.

**Turbulence**. For the measurement program with statistical turbulence, we investigate the variance, the turbulence intensity and the resolvable scales. From the PSD it can be observed that the spectra measured neither by the CTA nor by the UAS follow the Kolmogorov model over the entire frequency range (Fig. 8). Only in the high-frequency range from $10^1$ Hz and also in the low-frequency range for the $V(t)$ setup the spectra follow the Kolmogorov model. The transition range deviates in all measurements. This is not of any further concern within this study, since the consistency of the UAS with the CTA, not with




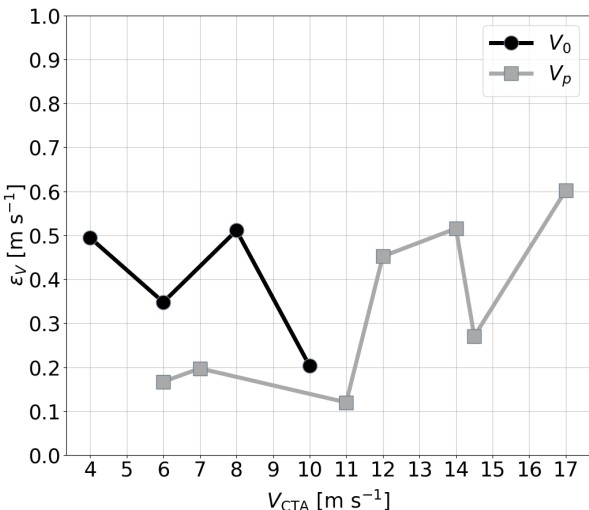

**Figure 6.** RMSE of UAS-based speed measurement $\epsilon_V$ in gusts over preset wind speed $V_{CTA}$. The different lines represent different speed sections of the time series, i.e. the initial wind speed $V_0$ and maximum wind speed $V_p$.

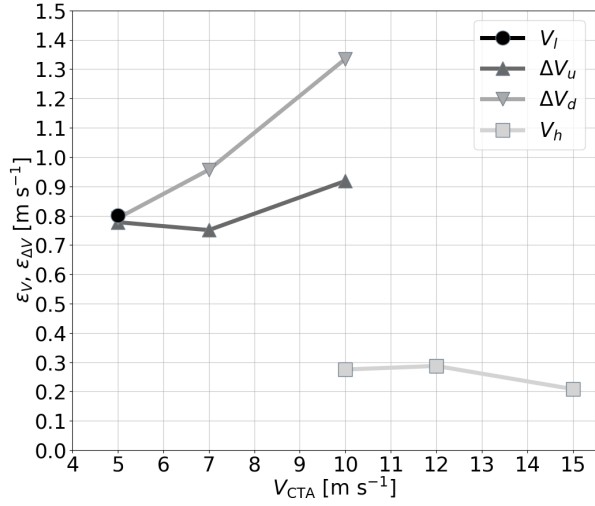

**Figure 7.** RMSEs of UAS-based speed measurements $\epsilon_V$ and $\epsilon_{\Delta V}$ in velocity steps over preset wind speed $V_{CTA}$. The different lines represent different speed sections of the time series, i.e the measured higher stationary wind speed $V_h$, the lower stationary wind speed $V_l$, the upward velocity step $\Delta V_u$ and the downward velocity step $\Delta V_d$.



the model, is investigated. In general, the spectra of the UAS and CTA measurements agree well, the $R^2$-value of 0.886 at a
$p$-value of $2 \cdot 10^{-10}$ show a very strong correlation between the variances $\sigma_{\mathrm{UAS}}^2$ and $\sigma_{\mathrm{CTA}}^2$. However, if the variance of the

265 turbulence is below approx. $0.5 \, \mathrm{m}^2 \, \mathrm{s}^{-2}$, the spectra diverge at about 2 Hz, and for $0.3 \, \mathrm{m}^2 \, \mathrm{s}^{-2}$ and below, there is a divergence
at about 1 Hz. This fact can also be identified in Fig. 9: The correlation between CTA and UAS measurements changes below
$0.5 \, \mathrm{m}^2 \, \mathrm{s}^{-2}$ and differs strongly below $0.3 \, \mathrm{m}^2 \, \mathrm{s}^{-2}$ compared to the higher variance ranges. Furthermore, a bias in the form of
a systematic underestimation of the UAS values can be observed. This occurs when the variance of the reference measurement
is determined over all frequencies up to the maximum frequency $f_{\mathrm{m,CTA}}$ that can be resolved by the CTA. However, if the

270 variance of the CTA is only calculated up to the maximum frequency $f_{\mathrm{m,UAS}}$ measured by the UAS, this bias does not occur.
This fact is also reflected in the RMSE for measurement of the turbulence intensity (see Fig. 10): $\epsilon_I$ increases significantly
at a variance of less than $0.3 \, \mathrm{m}^2 \, \mathrm{s}^{-2}$. This effect is due to the signal to noise ratio of the accelerometer measurements and
self-induced vibrations by the UAS which are stronger than the atmospheric flow signal below certain energy levels.

## 5   Discussion

Our experiments show that for measurements carried out using UAS of the SWUF-3D fleet, the RMSE of speed determination
along the wind direction exceeds $0.5 \, \mathrm{m \, s^{-1}}$ only in ranges that are close to the extrema of accelerations and wind speeds.
Generally, the determined accuracy of the UAS measurements has to be assessed considering the uncertainty of wind speeds
in the wind tunnel test section as described in Sect. 2.2 and the uncertainties of the CTAs mentioned in Sect. 2.3.

For the wind measurements we observe an accuracy which is comparable to the results in other studies: González-Rocha

et al. (2023) obtain an RMSE of $0.6 \, \mathrm{m \, s^{-1}}$ for time averages of about $2 \, \mathrm{min}$ intervals in hover flight and an RMSE of $2.5 \, \mathrm{m \, s^{-1}}$
in ascending vertical flight, and in their comparisons for the rigid body method González-Rocha et al. (2019) find an RMSE of
0.4 to $0.7 \, \mathrm{m \, s^{-1}}$. When time-averaged over $20 \, \mathrm{s}$, in the measurements of Neumann and Bartholmai (2015) the RMSE for the
wind speeds in an outdoor flight are $0.6 \, \mathrm{m \, s^{-1}}$ for hover and $0.36 \, \mathrm{m \, s^{-1}}$ for motion flight, in the wind tunnel experiments of
Neumann et al. (2012) the RMSE is $0.6 \, \mathrm{m \, s^{-1}}$. It should be emphasized that Neumann and Bartholmai (2015) measured at AoS

of $0°$, $45°$and $90°$ to the main flow direction and up to $8 \, \mathrm{m \, s^{-1}}$ wind speed, and found that there are no significant differences
in the wind measurement due to the heading of their system. In contrast to this, in our investigations within this speed range,
the accuracy at different AoS are sufficiently low, but differ significantly for various yaw angles. Brosy et al. (2017) obtained
an RMSE of $0.3 \, \mathrm{m \, s^{-1}}$ in free-field calibration of their system up to about $6 \, \mathrm{m \, s^{-1}}$ wind speed and an RMSE of $0.7 \, \mathrm{m \, s^{-1}}$ in
time series smoothed with a $10 \, \mathrm{s}$ moving average in flight tests. Also using $10 \, \mathrm{s}$ periods, the RMSE for the measurements by

290 Palomaki et al. (2017) is $0.32 \pm 0.17 \, \mathrm{m \, s^{-1}}$. Simma et al. (2020) achieve an RMSE between 0.26 and $0.29 \, \mathrm{m \, s^{-1}}$ for horizontal
wind estimates. The RMSE in the measurements of Segales et al. (2020) is $0.77 \, \mathrm{m \, s^{-1}}$. Of particular interest is that Wetz and
Wildmann (2023) report an RMSE of $0.25 \, \mathrm{m \, s^{-1}}$ for the mean wind speed with the calibration coefficients from Wetz et al.
(2021), since here measurements are performed with the same setup like in our investigations but with the coefficients from a
calibration in the free field. However, when applied to the validation data from the wind tunnel, these calibration values have a

295 significantly higher error than the calibration coefficients determined in the wind tunnel. Also, the measurements in Wetz and





**Figure 8.** Comparison of exemplary PSD of the wind speed $S$ determined via UAS and CTA measurements at different preset wind speeds $V$ and turbulence intensities $I$. The dotted grey line represents the spectrum following the Kolmogorov $-5/3$ power law, the black line represents the PSD derived from CTA measurements and the red line represents the PSD derived from UAS measurements. The spectra are processed with bin averages in the frequency space to decrease the noise.



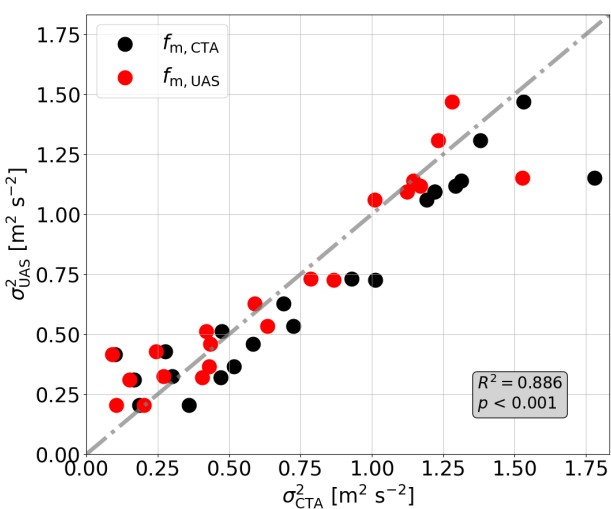

**Figure 9.** Comparison of the variances $\sigma^2$ obtained via the measurements of UAS and CTA, with $\sigma^2_{\mathrm{CTA}}$ determined with the measurement frequency of the CTA $f_{\mathrm{m,CTA}}$ (black dots) and with the measurement frequency of the UAS $f_{\mathrm{m,UAS}}$ (red dots) as maximum frequency.

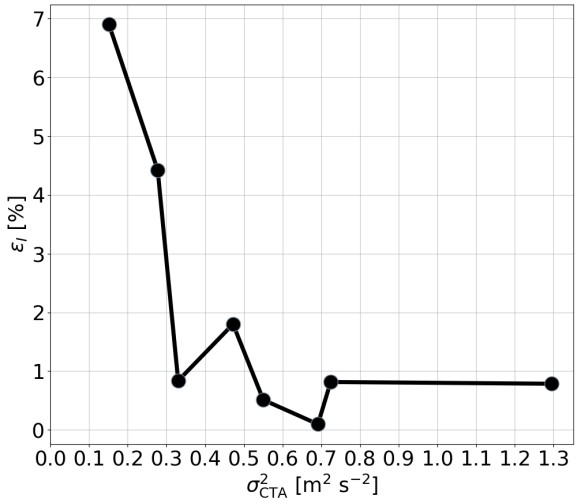

**Figure 10.** RMSE of UAS-based measurement of turbulence intensity $\epsilon_I$ over variance $\sigma^2_{\mathrm{CTA}}$ determined with the CTA.



Wildmann (2023) are performed in mean wind speed of 6 to $12.5 \mathrm{~m~s}^{-1}$. Both in the former and in the new calibration, the measurement uncertainties are lowest in this speed range. The measurements in the wind tunnel, in contrast, have been carried out at significantly lower and higher wind speeds, which contributes to the overall increased RMSE. Nevertheless, there is agreement in the turbulence measurement: Wetz and Wildmann (2022) also identify the variance of $0.3 \mathrm{~m}^2 \mathrm{~s}^{-2}$ and 2 Hz as

lower limit of resolvable turbulence and temporal resolution. Deviations from the reference in the turbulence measurement at the higher frequencies are also found by Shelekhov et al. (2022), but not as pronounced as in the cases presented here.

In general, our measurement setup exhibits some deficiencies: The optical flow sensor that was used does not maintain the position without error, but causes a slight movement in the horizontal plane. This leads to uncertainties, as the movement affects the wind measurements and cannot be compensated due to the unavailability of GNSS. Since the sensor itself could be

narrowed down as the source of this error, the use of a different optical flow system can potentially eliminate this inaccuracy. Some of the measurements carried out could not be used for the analyses in this paper as errors in the CTA data occurred.

We found throughout the process of calibration that different calibration coefficients for different velocity ranges in the longitudinal direction achieve higher accuracy for the wind speed measurements. This and applying coefficients from the lateral calibration lead to good results at different AoS. The accuracy at $0°$ AoS is best and similarly good at $10°$, while measurement

errors at AoS from $20\ldots60°$ are higher than desired, especially at high wind speeds. This shows that it is appropriate and essential to operate in weather vane mode. Analyses of deviations between UAS heading and measured wind direction of actual field measurements as presented in Wetz and Wildmann (2023) show that the directional error is always below $30°$ for velocities from $2 \mathrm{~m~s}^{-1}$ and below $20°$ for wind speeds above $4 \mathrm{~m~s}^{-1}$, and generally decreases with higher wind speeds. This means that AoS occurring at different wind speeds and inaccuracies in the wind speed measurement as a result of AoS

counteract each other. The calibration coefficients obtained are portable to other UAS in the fleet without a significant loss in accuracy. The already mentioned single outlier in the calibration of UAS no. $34$ is not caused by putative limitations of the portability of coefficients but by a random experimental error during that flight.

When measuring gusts, the RMSE of the stationary speeds is slightly above the RMSE of calibration flights. During these flight phases, the position corrections were carried out, and their impact on the wind measurement cannot be corrected com-

pletely. Also, the RMSE for very high velocities at the peak of the gust profile is slightly above that of calibration flights. The RMSE for the time response of $4.5 \cdot 10^{-1}$ s is satisfactory in this regard, especially with respect to the standard deviation of the CTA of $3.6 \cdot 10^{-1}$ s, the inhomogenity in the wind field as well as the distortions of the experienced gust duration at the UAS due to its motion. However, due to the positional drift, these results should be taken with some caution, as the position of the UAS with respect to the CTA may have changed within the considered time frame, which may also distort the timescale in

which gust characteristics reach the UAS.

In the same manner as for the gusts, the RMSE of the stationary speeds is found to be higher for the velocity steps than for calibration flights, which is also due to position corrections during these phases. The error in the upward and downward velocity steps is high, but also within limits when this is related to the flow condition: at a velocity drop of $10 \mathrm{~m~s}^{-1}$ within approx. 1.5 s the RMSE of approx. $1.3 \mathrm{~m~s}^{-1}$ is still acceptable for our applications.





For the turbulence measurement, the RMSE for the wind speed is slightly higher than for the calibration flights due to the positional drift. As mentioned above, the turbulence spectra measured by CTA and UAS both deviate from the Kolmogorov $-5/3$ power law and exhibit a characteristic hump at approximately 1 Hz. This is of minor importance for the analyses of this work, merely the agreement of the spectra is of relevance as well as accuracy in measuring turbulence intensity. This is found to be satisfactory for variances higher than $0.5 \text{ m}^2 \text{ s}^{-2}$, and for frequencies up to 2 Hz at variances above $0.3 \text{ m}^2 \text{ s}^{-2}$.

A systematic bias between variances obtained via UAS and CTA occurs which is caused by the different frequency resolution and the missing energy of the small scales that cannot be measured with the UAS. Nevertheless, the very strong correlation between the measured variances of UAS and CTA indicate a consistent measurement quality across a major range of variances. However, the goal is still to resolve smaller-scale turbulence, which probably requires a smaller airframe and smaller propellers for the UAS, which in turn is associated with lower MTOW and thus smaller batteries, reducing the maximum flight time.

Nonetheless, due to the dataset and the properties of the turbulent flow measured in this work, additional turbulence measurements are necessary in future to enable a more detailed quantification of the turbulence estimation capabilities of the SWUF-3D fleet. This requires the generation of statistical turbulence with PSDs not exhibiting the aforementioned hump but with a closer agreement with the $-5/3$ slope, or at least some distinct characteristic slope which is clearly discernible from white noise. This is essential in order to verify a potentially more general measurable minimum also at those frequency ranges. This cannot be

definitively deduced from the data available so far.

## 6 Conclusions

In this work, we successfully performed an advanced calibration of our wind measurement using UAS under laboratory conditions in a wind tunnel. Using the possibilities of the active grid of the wind tunnel, we were able to verify the accuracy of the wind measurement using the obtained new calibration coefficients in a broad variety of flow conditions.

By means of the measurement accuracies at different AoS in steady wind speeds we have demonstrated the necessity of the weather vane mode for our system and showed that measurements are reliable even in case of delayed response or errors of the yaw control. Without any angle of sideslip, the uncertainty remains below $0.4 \text{ m s}^{-1}$, but as soon as AoS increases above $20°$, a significant increase in uncertainty is observed. Especially above $9 \text{ m s}^{-1}$ errors of more than $0.7 \text{ m s}^{-1}$ are found.

By determining the measurement accuracy using the identical calibration coefficients on other UAS in the fleet, we were

able to demonstrate the portability of these coefficients to equivalent systems. For the measurements of all evaluated UAS, the RMSE is below $0.5 \text{ m s}^{-1}$. This allows for further up-scaling of the fleet without the need to perform individual wind tunnel calibrations on new UAS of this type.

Based on the different measurement scenarios using the active grid, we were able to verify the measurement accuracy in more complex and dynamic flow conditions. For measurements in gusts, the maximum RMSE of about $0.6 \text{ m s}^{-1}$ was

obtained at $17 \text{ m s}^{-1}$ wind speed. Furthermore, steady winds were also measured directly after abrupt velocity steps with an acceptable error (the respective lower velocities with an error of $0.8 \text{ m s}^{-1}$ and the upper ones with less than $0.3 \text{ m s}^{-1}$). For the measurements of the extreme wind speed changes, we observed a significant increase in the RMSE for the speed



measurement. Here, the RMSE for the response time is $7.1 \cdot 10^{-1}$ s and for the gust time measurement it is $4.5 \cdot 10^{-1}$ s. On the basis of measurements in flows with statistical turbulence, we were able to detect limits for the resolution of turbulent winds.

Generally, we achieve a high level of agreement in the measurement of turbulence parameters between UAS and reference CTA, but we observed a significant increase of the measurement errors for absolute variances below $0.3$ m$^2$ s$^{-2}$. In this variance range, the PSD of UAS and CTA diverge at frequencies above approx. 1 Hz. Between $0.3$ m$^2$ s$^{-2}$ and $0.5$ m$^2$ s$^{-2}$, this maximum frequency for accurate turbulence measurement is about 2 Hz, and no systematic divergence has been found at higher variances. However, due to the properties of the turbulence spectra of generated flows by the active grid, there remains

a need to investigate on further limits of turbulence resolution in terms of variance and frequency and their interaction, both within and beyond the value ranges considered in this work.

Summarizing, calibration along the main axes of the UAS is not fully sufficient in all inflow directions without the weather vane mode. Based on the validation with other UAS of the fleet, the hypothesis of portability could be confirmed. The validation in gusts, velocity steps and statistical turbulence showed that with the calibration method in stationary wind speeds it is possible

to reliably measure highly dynamic flows with the UAS. Further studies with more extensive and detailed turbulence analysis in the wind tunnel will also include other airframes with different rotor sizes. In addition, there is a need to integrate the vertical wind component (Wildmann and Wetz, 2022) into the type of analysis presented in this paper, ideally along with calibration under similar laboratory conditions. Since the work presented here is part of the SWUF-3D fleet research, it is important to integrate the established accuracy into the analysis of uncertainties in spatially distributed fleet measurements, which needs to

be validated via measurements in field experiments.



# A Verification measurement presets

**Table A1.** Measurement presets with various AoS

| $|AoS|$ [°] | $V_{\max}$ [m s$^{-1}$] | $T_s$ [s] | $r$ [$-$] |
|---|---|---|---|
| 0 | 16 | 60 | 3 |
|   | 18 | 30 | 1 |
|   |    | 60 | 2 |
| 10 | 16 | 30 | 2 |
|    | 18 | 60 | 2 |
| 20 | 16 | 30 | 2 |
|    |    | 60 | 1 |
|    | 18 | 60 | 1 |
| 30 | 16 | 30 | 2 |
|    |    | 60 | 1 |
|    | 18 | 60 | 1 |
| 60 | 16 | 30 | 2 |
| 90 | 16 | 30 | 2 |
|    |    | 60 | 1 |
|    | 18 | 60 | 1 |





**Table A2.** Measurement presets for gusts

| $V_0$ [m s$^{-1}$] | $V_g$ [m s$^{-1}$] |
|---|---|
| 4.0 | 2.0 |
| | 3.0 |
| 6.0 | 5.0 |
| 8.0 | 4.0 |
| | 6.0 |
| 10.0 | 4.5 |
| | 7.0 |





**Table A3.** Measurement presets for velocity steps

| $V_0$ [m s$^{-1}$] | $V_{\mathrm{du}}$ [m s$^{-1}$] |
|---|---|
| 5.0 | 5.0 |
| | 7.0 |
| | 10.0 |



**Table A4.** Measurement presets for statistical turbulence

| $V_0$ [m s$^{-1}$] | $I$ [%] | $r$ [$-$] |
|---|---|---|
| 5.0 | 11.5 | 2 |
| 7.0 | 11.5 | 2 |
| 9.0 | 11.5 | 2 |
| 11.0 | 5.5 | 1 |
|  | 11.5 | 2 |
| 13.0 | 11.5 | 2 |
| 15.0 | 5.5 | 1 |
|  | 11.5 | 2 |
| 17.0 | 11.5 | 2 |
| $V(t)$ | 9.5 | 2 |
|  | 15.0 | 3 |



## B Nomenclature

| | |
|---|---|
| ABL | atmospheric boundary layer |
| AoS | angle(s) of sideslip |
| $b_x$, $b_y$ | exponential calibration coefficients |
| $c_x$, $c_y$ | linear calibration coefficients |
| CTA | Constant Temperature Anemometer |
| $f$ | frequency |
| $F_w$ | wind force |
| GNSS | global navigation satellite system |
| $I$ | turbulence intensity |
| PSD | power spectral density |
| $r$ | number of runs |
| **R** | rotation matrix |
| RMSE | root-mean-square error |
| $S$ | PSD |
| $t$ | time |
| $T_s$ | time for which wind speeds are kept constant for calibration |
| $u$ | longitudinal wind speed component |
| UAS | unmanned aerial system |
| $v$ | lateral wind speed component |
| **V** | wind speed vector |
| $V_0$ | inertial wind speed of gust, lower wind speed of velocity step and fan wind speed for statistical turbulence aimed for |
| $V_{du}$ | wind speed aimed for of upward and downward velocity step |
| $V_g$ | targeted gust velocity amplitude |
| $V_h$ | higher wind speed between velocity steps |
| $V_l$ | lower wind speed between velocity steps |
| $V_p$ | maximum speed in gust passage |
| $\dot{x}_{gps}$ | velocity in longitudinal direction of the geodetic coordinate system detected by the GNSS |
| $\dot{y}_{gps}$ | velocity in lateral direction of the geodetic coordinate system detected by the GNSS |
| | |
| $\Delta T_d$ | duration of downward velocity step |
| $\Delta T_g$ | duration of gust between downward peaks |
| $\Delta T_u$ | duration of upward velocity step |



| $\Delta V_d$ | wind speed of downward velocity step and gust |
| $\Delta V_u$ | wind speed of upward velocity step and gust |
| $\epsilon_I$ | RMSE of turbulence intensity |
| $\epsilon_V$ | RMSE of wind speed |
| $\epsilon_{\Delta T}$ | RMSE of time deltas |
| $\epsilon_{\Delta V}$ | RMSE of wind speed deltas |
| $\theta$ | pitch angle |
| $\sigma$ | standard deviation |
| $\sigma^2$ | variance |
| $\phi$ | roll angle |
| $\psi$ | yaw angle |



*Data availability.* The data is available under doi.org/10.5281/zenodo.10047738

*Author contributions.* JK wrote the main manuscript and performed the calibrations and data analysis. The experiment was conducted by
JK. NW provided the algorithm to measure wind with the quadrotor, was involved in the experiment and contributed to the manuscript. LN
provided the wind tunnel setups and wrote the description of the wind tunnel.

*Competing interests.* The authors declare that they have no conflict of interest.

*Disclaimer.* This research is part of the project ESTABLIS-UAS and has been supported by the HORIZON EUROPE European Research
Council (grant no. 101040823).
The article processing charges for this open access publication were covered by the German Aerospace Center (DLR).

*Acknowledgements.* We thank Tamino Wetz and Michael Hölling for their assistance during the wind tunnel measurements. Andreas Marsing
internally reviewed the manuscript and we thank him for his valuable comments. Pixhawk is a trademark of Lorenz Meier.



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
