# Peer review of "High resolution wind speed measurements with quadcopter UAS: calibration and verification in a wind tunnel with active grid"

_EGUsphere, 2024_

## Referee Comment (RC1)

1. Does the paper address relevant scientific questions within the scope of AMT? **Yes**

2. Does the paper present novel concepts, ideas, tools, or data? **Novel tools**

3. Are substantial conclusions reached? **Yes, but needs some more work**

4. Are the scientific methods and assumptions valid and clearly outlined? **Yes**

5. Are the results sufficient to support the interpretations and conclusions? **Yes in part**

6. Is the description of experiments and calculations sufficiently complete and precise to allow their reproduction by fellow scientists (traceability of results)? **Yes**

7. Do the authors give proper credit to related work and clearly indicate their own new/original contribution? **Yes**

8. Does the title clearly reflect the contents of the paper? **Yes**

9. Does the abstract provide a concise and complete summary? **Yes (minor corrections below)**

10. Is the overall presentation well structured and clear? **Yes**

11.

12. Is the language fluent and precise? **Needs some grammar and narrative corrections.**

13. Are mathematical formulae, symbols, abbreviations, and units correctly defined and used? **Yes**

14. Should any parts of the paper (text, formulae, figures, tables) be clarified, reduced, combined, or eliminated? **Format of the tables and some citations need change.**

15. Are the number and quality of references appropriate? **Yes**

16. Is the amount and quality of supplementary material appropriate? **Yes**

**General comments:**

The manuscript describes the an approach to measuring wind speeds using a quadcopter UAS within a wind tunnel, emphasizing calibration and verification methods. The authors are looking to refine the calibration process for the wind measurement algorithm of their SWUF-3D UAS fleet within a controlled laboratory setting. This process is important for obtaining accurate in situ measurements of the atmosphere without waiting for favorable weather conditions for a proper calibration in the open field. Other researchers on the topic have been discussing this method to be a simple and obvious solution, but efforts must be made to connect this laboratory studies with the real scenarios in the open field. Overall I find the paper well structured and I'm happy to see this method providing a more robust wind calibration. However, the authors may need to correct some narrative to facilitate the reader to follow the study and other concerns which I discuss in more detail below. This concerns and others need to be addressed before I can recommend publication in AMT.

**Specific comments:**

In the abstract, it is not clear what is being calibrated. The authors mention "the algorithm for wind measurement" and "calibration coefficients" but this is not very specific. Is it for an onboard instrument? Is it for a dynamic model? Is it for drone's autopilot system? Consider making it clear since this is where the reader gets their first impression.

Line 91-93: Please specify if you are using raw GNSS measurements for the correction. If this is the case, why not use the fused solution given by the IMU? Besides providing higher sampling rate, it should also be more precise since it is correcting the GNSS data with the accelerometers, barometers, and gyros. This also applies to the Optical Flow and rangefinder devices. Also, Optical flow works best when there are clear patterns on the floor. Have you tried painting or drawing patterns/grids on the floor to help increase the accuracy of the optical flow and decrease the drift?

Line 100: Have you considered using an indoor GPS repeater? This may allow you to reproduce similar positioning conditions as flying in the free field.

The way Table A1 is laid out is hard to understand especially with the empty cells.

Calibration section 3.1: I'm assuming that the drone was calibrated with the turbulent wind tunnel set to laminar flow as much as possible. Please clarify this in the experimental description.

Line 172-174: This is a statement given by the authors without much reasoning. Please show an equation or deduction where the accelerometer offsets are the only uncommon factor among the equally-build UASs for wind estimation. Also, the authors claim no wind tunnel is required. However, the way I understand this is that at least 1 drone needs wind tunnel calibration and then the rest of the fleet would get the coefficient by portability. Please clarify if this is the case.

Angles of sideslip section: If I understood well, the authors mean that the slow response of the weathervane function is not able to capture/resolve small-scale turbulence. For this reason, there are lateral perturbations not being considered in the wind estimation. Therefore, authors seek to determine these errors by manually adjusting the AoS and study the behavior in a wind tunnel. If this is correct, then why relevance only at low wind speeds? Is the intention here to just measure the error or to also correct for it?

Line 215: What do the authors mean by timing accuracy? To me it looks like Eq.5 is taking the RMSE of the time response between the UAS and CTA.

Turbulence section: Were the measurements taken with both the UAS and CTAs running at the same time or one at the time? I can imagine that the turbulent wake of the UAS will severely impact the CTA measurements, especially for the PSD. Please clarify the measurement process for the turbulence study.

Discussion section: Even though the authors saw some position drift using the optical flow, the optical flow should be more accurate than a GNSS system for a large margin. This alone could have been a contributor to the lower overall RMSE shown by the authors. However, by removing the GNSS uncertainties, the author's calibration coefficients and results may be more representative of the UAS's geometry, autopilot response, and propulsion system. This is a valid and usable result but I'm afraid it is not fair to compare these results with the drones set up for the open field, at least not in a straightforward manner. The authors should make an effort to discuss or investigate a way to

translate this results (maybe using the optical flow and GNSS uncertainties as proxy) if the goal is to use this technique for wind measurements in the planetary boundary layer where a GNSS is most commonly used. If the position error of the Optical Flow is similar to the GNSS, then the comparison can be deemed as fair but please state it on the text.

**Technical corrections:**

Brosy et al. 2017 – wrong DOI

Line 26: consider replacing "steady flight" with "vertical profile" or similar, since steady flight implies hover too.

Line 60-61: Odd sentence, please reword it.

Line 78: "from axis to axis" is not clear what axis. Do you mean the diagonal from rotor to rotor?

Table 1: Capitalize first letter of each first word on the list.

Line 81: Consider replacing with "Wind measurements are taken by hovering the UAS in one place …"

Line 84-86: Consider replacing with "The wind acting on the UAS during hover can be determined by applying the wind algorithm using the modified Rayleigh drag equation in Eq. 1 (citation) to the measured attitude and …"

Line 101: "discernible systematic" reads wrong. Please correct.

Line 148-150: Sentence reads wrong. Consider replacing with "To compute the transfer function … in the modified Rayleigh drag equation (Eq.1) that best fit …"

Line 160: replace coordinate directions with axes.

Line 168 and 174: citations not in the right format

Line 182: "30 s resp. 60 s" is confusing.

Line 190: the expression can be directly written $V_p = V_o + V_g$

Line 275: RMSE of "wind" speed. Remove determination

There are several grammatical and syntax errors throughout the paper. Although most of it can be understood, some readers may find it difficult to follow. I strongly recommend the authors revisit the narrative of the paper and even use grammar/spell checkers if needed.

---

## Referee Comment (RC2)

**General Comments:**

The paper entitled, 'High resolution wind speed measurements with quadcopter UAS: calibration and verification in a wind tunnel with active grid' presents the validation results of UAS-based wind estimates obtained by performing flight experiments in an open-section wind tunnel with an active grid. The UAS wind estimation performance was assessed by varying flow conditions and the aircrafts sideslip angle. This work is important to understand the reliability of UAS in measuring wind speed and turbulence within the planetary boundary layer. However, the authors need to address the following points before I can recommend publication in AMT.

**Specific Comments:**

Line 1: The manuscript states, "As a contribution to closing observational gaps in the atmospheric boundary layer (ABL), the SWUF-3D fleet of unmanned aerial systems (UAS) is utilized for in situ measurements of turbulence." Here it would be helpful to tell the reader what scales of turbulence is the SWUF-3D platform able to resolve.

Line 2: The manuscript states, "To date, the algorithm for wind measurement has only been calibrated in the free field." Here the authors need to specify which specific algorithm they are referring to. Additionally, it's unclear if the authors are using the words 'turbulence measurement' and 'wind measurement' interchangeably. If not, and therefore the authors need to make the distinction between turbulence and wind velocity measurement with more clarity.

Line 10: The manuscript states, "our analyses show that the uncertainty depends on the wind speed magnitude and increases with higher wind speeds, resulting in an overall root-mean squared error (RMSE) of less the 0.2 m s$^{-1}$." However, it is not explicitly stated which type of uncertainty the authors are referring to.

Line 12: The manuscript states, "The maximal RMSE occurs in the most extreme velocity steps (i.e., a lower speed of 5 m s$^{-1}$ and an amplitude of 10 m s$^{-1}$) and exceeds 1.3 m s$^{-1}$. This result seems to contradict the result reported in Line 10.

Figure 1: It would be helpful for the authors to denote the distance between the points a, b, and c, as well as the position of all 7 CTAs and the Prandtl probe in Figure 1. Additionally, since the calibration experiments were performed in a wind tunnel with an open test section, were any experiments performed to quantify the wind field differences across points b and c?

Line 100: The positional drift should be reported in units of distance (i.e., m) instead of units of speed (i.e., m s$^{-1}$).

Line 101: The manuscript states, "As wind speed increases, the intensity and direction of the drift change without a discernible systematic, which required constant adjustment counteracting the drift during the test flights. These adjustments were executed by the remote pilot through a manual trim." It would be useful for the reader to know if the manual trim remained constant across all test cases, and if any experiments were performed to quantify how the manual trimming affected the accuracy of wind estimates.

Line 123: The manuscript states, "Test runs with no UAS show that all CTAs measure the equivalent wind speed with sufficient accuracy: the standard deviation of the measured wind speed of the individual CTAs

is less than 0.05 m/s." Is there a figure showing these results? Why not use instead the absolute error or root mean squared error to compare the performance of CTAs? Additionally, did the authors perform an analysis to determine the error between the CTAs and the Prandtl probe?

Line 135: The manuscript states, "Careful quality checks were carried out for the CTA measurement data and corrupted data was sorted out." It would be useful for the reader to know the process or criteria that was used validate the quality of CTA measurement data.

Line 148: It would be useful for the reader to know which specific optimization algorithm was used to estimate the calibration coefficients for Eq. 1.

Line 150: A reference is needed for ISO 17713-1:2007

Table A1 is missing entries in column two

Table A2 is missing entries in column one

Table A4 is missing entries in column one

---

## Author Comment (AC1)

**High resolution wind speed measurements with quadcopter UAS: calibration and verification in a wind tunnel with active grid**

Johannes Kistner[1], Lars Neuhaus[2,3], and Norman Wildmann[1]

[1]Deutsches Zentrum für Luft- und Raumfahrt, Institut für Physik der Atmosphäre, Oberpfaffenhofen, Germany
[2]Carl von Ossietzky Universität Oldenburg, School of Mathematics and Science, Institute of Physics
[3]ForWind - Center for Wind Energy Research, Küpkersweg 70, 26129 Oldenburg, Germany

**Correspondence:** Johannes Kistner (johannes.kistner@dlr.de)

We want to thank the reviewers for their careful and valuable review. We hope that we can clarify our analyses and clear out some of the concerns with our response.

**1 Review Comment 1**

**1.1 General comments**

5    1. *The manuscript describes the an approach to measuring wind speeds using a quadcopter UAS within a wind tunnel, emphasizing calibration and verification methods. The authors are looking to refine the calibration process for the wind measurement algorithm of their SWUF-3D UAS fleet within a controlled laboratory setting. This process is important for obtaining accurate in situ measurements of the atmosphere without waiting for favorable weather conditions for a proper calibration in the open field. Other researchers on the topic have been discussing this method to be a simple and*

10    *obvious solution, but efforts must be made to connect this laboratory studies with the real scenarios in the open field. Overall I find the paper well structured and I'm happy to see this method providing a more robust wind calibration. However, the authors may need to correct some narrative to facilitate the reader to follow the study and other concerns which I discuss in more detail below. This concerns and others need to be addressed before I can recommend publication in AMT.*

15    Thank you very much for the positive feedback, we are also happy to achieve the more robust wind calibration. As part of our replies to the specific comments, we made efforts to connect our laboratory study with real scenarios in the open field and to better describe this connection. The narrative was reviewed internally by a native speaking scientist, and their corrections will be implemented in a revised manuscript. We hope that this will make it easier for the reader to understand our study.

20  ### 1.2 Specific comments

1. *In the abstract, it is not clear what is being calibrated. The authors mention "the algorithm for wind measurement" and "calibration coefficients" but this is not very specific. Is it for an onboard instrument? Is it for a dynamic model? Is it*

*for drone's autopilot system? Consider making it clear since this is where the reader gets their first impression.*

Our horizontal wind measurement is based on measuring the acceleration of the UAS using the avionics sensors to infer the wind acting on the UAS. This conversion is carried out using a wind measurement algorithm that is based on several transformation terms. These contain coefficients to be derived empirically, that we determine through wind tunnel tests, which we refer to as calibration of the wind measurement algorithm. We agree that the abstract does not contain a sufficiently detailed description and will use the following explanation in the abstract of a revised manuscript: "To date, the coefficients for the transformation terms used in our algorithm for deriving wind speeds from avionic data, have only been determined via calibration flights in the free field."

2. *Line 91-93: Please specify if you are using raw GNSS measurements for the correction. If this is the case, why not use the fused solution given by the IMU? Besides providing higher sampling rate, it should also be more precise since it is correcting the GNSS data with the accelerometers, barometers, and gyros. This also applies to the Optical Flow and rangefinder devices. Also, Optical flow works best when there are clear patterns on the floor. Have you tried painting or drawing patterns/grids on the floor to help increase the accuracy of the optical flow and decrease the drift?*

We actually use the fused solution given by the IMU and will specify this in the revised manuscript as we expect it to be more precise especially for small movements. In field measurements, deviations of approx. $0.1$ m s$^{-1}$ between raw and fused output data can occur. Also, the fused solution has the advantage that it is available at a higher frequency than the 5 Hz raw GNSS output. For troubleshooting the optical flow drift, we tried different surfaces, regular vs. irregular, rectilinear vs. organic ground patterns and different lighting conditions. The drift was not affected, let alone reduced.

3. *Line 100: Have you considered using an indoor GPS repeater? This may allow you to reproduce similar positioning conditions as flying in the free field.*

Yes, we considered using an indoor GPS repeater, but for practical reasons (effort of installation and cost) we decided in favor of optical flow.

4. *The way Table A1 is laid out is hard to understand especially with the empty cells.*

We will improve this for the revised manuscript.

5. *Calibration section 3.1: I'm assuming that the drone was calibrated with the turbulent wind tunnel set to laminar flow as much as possible. Please clarify this in the experimental description.*

Yes, this assumption is correct and we agree that this should be made clearer. In a revised manuscript, we will be more precise about this.

6. *Line 172-174: This is a statement given by the authors without much reasoning. Please show an equation or deduction where the accelerometer offsets are the only uncommon factor among the equally-build UASs for wind estimation. Also, the authors claim no wind tunnel is required. However, the way I understand this is that at least 1 drone needs wind tunnel calibration and then the rest of the fleet would get the coefficient by portability. Please clarify if this is the case.*

The UAS in the fleet have the same mechanical and aerodynamic properties and the center of gravity lies in the same

position for all UAS. The respective UAS therefore always has the same aerodynamic orientation in the wind, i.e. along the wind direction and with a corresponding pitch angle $\theta$, in order to maintain its position against the wind. The formulas for the horizontal winds were already described by Wetz and Wildmann (2022):

$$F_x = mg\sin(\theta) + m\ddot{x}$$

$$F_y = mg\cos(\theta)\sin(\phi) + m\ddot{y}$$

The individual components and their mounting in the UAS are not perfectly identical, but the deviations can be neglected in view of the minor influence on the aerodynamic and mechanical behavior of the UAS. For example, Wildmann (2022) was also able to show that several equally built rotors show little aerodynamic difference, even with small damages on the propellers. Only the slight deviations in the orientation of the autopilots in relation to the respective UAS frame into which they are mounted have a relevant influence on the wind measurement. This is not due to any aerodynamic or flight mechanical effects caused by the autopilot; those deviations are as negligible as in the case of the other components. However, the acceleration sensors on which our wind measurement is based are installed in the autopilot. Accelerometers are calibrated in a manual procedure for each UAS. These calibrations can also lead to individual biases. Therefore, the wind measurement is very sensitive to the accelerometer offsets, which is why the offsets are the only relevant uncommon factor between the equally-built UAS for wind estimation. With regard to the extent to which wind tunnel measurements are necessary for determining the offsets and the calibration coefficients, the reviewer understood it as we meant it: if portability is given the coefficients can be applied to the measurement data of the other UAS of the fleet. A wind tunnel is necessary, or at least helpful, to determine the basic aerodynamic calibration. What we claim is that no wind tunnel is necessary for the calibration of the installation-related offsets of the individual UAS.

7. *Angles of sideslip section: If I understood well, the authors mean that the slow response of the weathervane function is not able to capture/resolve small-scale turbulence. For this reason, there are lateral perturbations not being considered in the wind estimation. Therefore, authors seek to determine these errors by manually adjusting the AoS and study the behavior in a wind tunnel. If this is correct, then why relevance only at low wind speeds? Is the intention here to just measure the error or to also correct for it?*

The slow response takes too long to capture wind direction changes on small scales, i.e. weak and sudden lateral wind components. Therefore, we use the roll axis to measure the lateral wind component, which we have also calibrated in the wind tunnel measurements (see section 3.1). Lateral perturbations are therefore considered in the wind estimation. The wind vector is calculated according to Eq. 1 in the manuscript, which is based on Wetz et al. (2021) and Wetz and Wildmann (2022) (see Sect. 2.1 of the manuscript). In the wind tunnel, we validated how well the wind algorithm performs if the UAS is not perfectly aligned with the main wind direction (section 3.2.2). The purpose is therefore to measure the error that we would have in the wind measurement during AoS. For this purpose, we have deactivated the weather vane mode in order to have a definite angle of sideslip to the main wind direction that is not corrected by the weather vane mode.

[Figure]

**Figure 1.** Absolute mean angle of sideslip |AoS| over mean measured wind speed $V$ for 60-second segments of 209 drone flights with an average flight duration of over 8 min each.

90    At a considerable mean wind speed in the atmosphere, no rapid and large changes are expected in wind direction. The lateral wind component will be relatively small and decreasing until the UAS is facing the wind. As soon as the UAS is subject to side winds, it will correct and yaw. For these two reasons, the higher the wind speed and the greater the slip angle, the less likely is the occurrence. Therefore we stated "In this context, smaller angles and errors in the lower wind speeds are of particular relevance." By analysing the deviation between the heading of the UAS (measured by

95    the compass) and the wind direction (measured via the wind measurement algorithm with calibration coefficients as set before the wind tunnel tests) during field tests, we were able to verify this: In Fig. 1 it can be seen that larger AoS occur at lower wind speeds, and the majority of AoS is below $10°$. The directional error is below $30°$, and below $20°$ for higher wind speeds above $4 \, \text{m s}^{-1}$, and generally tends to decrease with higher wind speeds. This is roughly the opposite behavior to the RMSE at different AoS in Fig. 4 of the manuscript.

8. *Line 215: What do the authors mean by timing accuracy? To me it looks like Eq.5 is taking the RMSE of the time response between the UAS and CTA.*

This is correct, Eq. 5 is taking the RMSE of the difference in time response of UAS and CTA which is what we meant by timing accuracy. A more clear wording will be done for the revised manuscript.

9. *Turbulence section: Were the measurements taken with both the UAS and CTAs running at the same time or one at the time? I can imagine that the turbulent wake of the UAS will severely impact the CTA measurements, especially for the PSD. Please clarify the measurement process for the turbulence study.*

It is correct that the wake of the UAS would strongly influence the measurements of the CTAs at flight altitude and below, not only for the turbulence measurements. As we explain in Section 2.3, we therefore choose CTA no. 2 as a reference for all measurements, as it is undisturbed by the UAS and it is valid to use a sensor for wind measurement without measuring at the exact same altitude as the UAS. We also carried out turbulence measurements without a UAS in order to compare the measurements of the reference CTA for measurements with and without a UAS in order to check whether the results match and whether the procedure therefore is reasonable (see Fig. 2). Implicit in the explanations in Section 2.3 is that we carry out the reference measurement with the CTAs and the UAS measurements at the same time. We recognize that an explicit clarification should be included, which we will include in the revised manuscript.

10. *Discussion section: Even though the authors saw some position drift using the optical flow, the optical flow should be more accurate than a GNSS system for a large margin. This alone could have been a contributor to the lower overall RMSE shown by the authors. However, by removing the GNSS uncertainties, the author's calibration coefficients and results may be more representative of the UAS's geometry, autopilot response, and propulsion system. This is a valid and usable result but I'm afraid it is not fair to compare these results with the drones set up for the open field, at least not in a straightforward manner. The authors should make an effort to discuss or investigate a way to translate this results (maybe using the optical flow and GNSS uncertainties as proxy) if the goal is to use this technique for wind measurements in the planetary boundary layer where a GNSS is most commonly used. If the position error of the Optical Flow is similar to the GNSS, then the comparison can be deemed as fair but please state it on the text.*

We carried out measurements in the open field using the optical flow sensor for position control, but the GPS data was also logged. The drift was compensated in the same way as in the wind tunnel using trim; the UAS moved horizontally a few meters in all directions, but remained above the hover position on average. The plots below (Fig. 3, 4) show the comparison of the speeds that were logged for the optical flow and the GPS, in the north and east directions. Note that we use the raw GPS data in this case in order to obtain the GPS data without any bias caused by the optical flow measurement.

It can be seen that the measured speeds differ to some extent. It is not possible to tell from this which measurement is the more correct one. For this purpose, the velocities are integrated up to the position (see Fig. 5).

Both sensors overestimate the deviations of the UAS position from the hover position. However, while according to GPS the UAS moves on average within a limited range of $10^1$ m, according to optical flow the UAS has moved a

[Figure]

**Figure 2.** Comparison of PSD derived with the reference CTA during measurements with and without UAS flying upstream of the CTA cross.

distance in the dimension of $10^2$ m away from the hover position. This means that the GPS measurement, which is - opposing to the optical flow data - not fused with other sensors to increase accuracy, is considerably closer to the actual movement of the UAS. The time series of the measured speeds using GPS are therefore more plausible than those of optical flow, which appears to overestimate the movements of the UAS (which shows in the drift that we mentioned as a source of error). While we do not claim from this analysis that GPS would be significantly more accurate than optical flow, it is certainly reasonable to assume that the wind measurement would not be more accurate simply by operating under optical flow. Consequently, we consider a comparison of the wind tunnel measurements with the UAS set up for outdoor measurements as fair. As long as the flight behaviour and the observed movement of the UAS are similar, the measurement accuracy will also be similar. However, we agree with the reviewer that we should address this subject in the manuscript, which we will do in a revised version.

[Figure]

**Figure 3.** Time series of UAS speed in north direction measured via optical flow vs. via GPS.

**1.3   Technical comments**

1. *Brosy et al. 2017 – wrong DOI*

   We double checked the reference against the *How to cite* paragraph on the journal website, the DOI is correct. However, it can happen that when copying the DOI from the manuscript, the line number is also copied due to the line break within the DOI.

2. *Line 26: consider replacing "steady flight" with "vertical profile" or similar, since steady flight implies hover too.*

   We will change "steady flight" to "steady directional flight" to include both, steady vertical and horizontal flight.

[Figure]

**Figure 4.** Time series of UAS speed in east direction measured via optical flow vs. via GPS.

3. *Line 60-61: Odd sentence, please reword it.*

   The sentence was reworded to "We highlight that this is the first time that wind measurements have been performed with a multicopter UAS in the reproducible turbulent flow fields of a wind tunnel with an active grid."

4. *Line 78: "from axis to axis" is not clear what axis. Do you mean the diagonal from rotor to rotor*

   This is what we meant, we will change the text in the manuscript accordingly.

5. *Table 1: Capitalize first letter of each first word on the list.*

   We will change the text in the revised manuscript as suggested.

6. *Line 81: Consider replacing with "Wind measurements are taken by hovering the UAS in one place . . .*

   We will change the text in the revised manuscript as suggested.

[Figure]

**Figure 5.** UAS position relative to initial hover position derived via optical flow vs. via GPS

7. *Line 84-86: Consider replacing with "The wind acting on the UAS during hover can be determined by applying the wind algorithm using the modified Rayleigh drag equation in Eq. 1 (citation) to the measured attitude and ..."*

   We will change the text in the revised manuscript as suggested.

8. *Line 101: "discernible systematic" reads wrong. Please correct.*

   We will change "discernible systematic" to "identifiable pattern".

9. *Line 148-150: Sentence reads wrong. Consider replacing with "To compute the transfer function ... in the modified Rayleigh drag equation (Eq.1) that best fit ..."*

   We will change the text in the revised manuscript as suggested.

10. *Line 160: replace coordinate directions with axes.*

    We will change the text in the revised manuscript as suggested.

170   11. *Line 168 and 174: citations not in the right format*

The citations will be corrected to the right format.

12. *Line 182: "30 s resp. 60 s" is confusing.*

We will change "30 s resp. 60 s" to "30 s or 60 s".

13. *Line 190: the expression can be directly written Vp = Vo + Vg*

175   We will change the text in the revised manuscript as suggested.

14. *Line 275: RMSE of "wind" speed. Remove determination*

We will change the text in the revised manuscript as suggested.

15. *There are several grammatical and syntax errors throughout the paper. Although most of it can be understood, some readers may find it difficult to follow. I strongly recommend the authors revisit the narrative of the paper and even use grammar/spell checkers if needed.*

180   We apologize for any trouble understanding our text. Grammar, spelling and the narrative were checked internally by a native speaker. The corrections will be implemented in a revised manuscript. Remaining errors are hopefully caught by the professional copy-editors of the Copernicus publisher.

**References**

185 Wetz, T., Wildmann, N., and Beyrich, F.: Distributed wind measurements with multiple quadrotor unmanned aerial vehicles in the atmospheric boundary layer, Atmospheric Measurement Techniques, 14, 3795–3814, https://doi.org/10.5194/amt-14-3795-2021 , 2021.

Wetz, T. and Wildmann, N.: Spatially distributed and simultaneous wind measurements with a fleet of small quadrotor UAS, Journal of Physics: Conference Series, 2265, 022 086, https://doi.org/10.1088/1742-6596/2265/2/022086, 2022.

Wildmann, N. and Wetz, T.: Towards vertical wind and turbulent flux estimation with multicopter UAS, Atmospheric Measurement Tech-
190 niques, 15, 5465—5477, https://doi.org/10.5194/amt-15-5465-2022, 2022.

---

## Author Comment (AC2)

**High resolution wind speed measurements with quadcopter UAS: calibration and verification in a wind tunnel with active grid**

Johannes Kistner[1], Lars Neuhaus[2,3], and Norman Wildmann[1]

[1]Deutsches Zentrum für Luft- und Raumfahrt, Institut für Physik der Atmosphäre, Oberpfaffenhofen, Germany
[2]Carl von Ossietzky Universität Oldenburg, School of Mathematics and Science, Institute of Physics
[3]ForWind - Center for Wind Energy Research, Küpkersweg 70, 26129 Oldenburg, Germany

**Correspondence:** Johannes Kistner (johannes.kistner@dlr.de)

We want to thank the reviewers for their careful and valuable review. We hope that we can clarify our analyses and clear out some of the concerns with our response.

**1 Review Comment 2**

**1.1 General comments**

5    1. *The paper entitled, 'High resolution wind speed measurements with quadcopter UAS: calibration and verification in a wind tunnel with active grid' presents the validation results of UAS-based wind estimates obtained by performing flight experiments in an open-section wind tunnel with an active grid. The UAS wind estimation performance was assessed by varying flow conditions and the aircratis sideslip angle. This work is important to understand the reliability of UAS in measuring wind speed and turbulence within the planetary boundary layer. However, the authors need to address the*
10    *following points before I can recommend publication in AMT.*

Many thanks for the positive feedback and the acknowledgement of our research's importance for turbulence measurement in the boundary layer. We hope that we can address the concerns satisfactorily with our answers below.

**1.2 Specific comments**

1. *Line 1: The manuscript states, "As a contribution to closing observational gaps in the atmospheric boundary layer*
15    *(ABL), the SWUF-3D fleet of unmanned aerial systems (UAS) is utilized for in situ measurements of turbulence." Here it would be helpful to tell the reader what scales of turbulence is the SWUF-3D platform able to resolve.*

Resolvable length scales can be found in Table 2 in Wetz et al. (2023), where fleet measurements are discussed in more detail. The largest resolvable length scales depend on the maximum flight and thus measurement duration, which in turn depends on atmospheric conditions. The smallest resolvable length scales depend on the maximum measurable
20    frequency, which also depends on the flow conditions, which is a subject of the manuscript. A benchmark here is that scales in the order of magnitude down to approx. 5 m can still be resolved. We find it questionable whether these explanations should be added to the content in line 1, especially since the focus of the manuscript is not on this aspect of

turbulence measurement. Nevertheless we agree that it would be helpful to the reader, and therefore mention this in the introduction of the revised manuscript.

2. *Line 2: The manuscript states, "To date, the algorithm for wind measurement has only been calibrated in the free field." Here the authors need to specify which specific algorithm they are referring to. Additionally, it's unclear if the authors are using the words 'turbulence measurement' and 'wind measurement' interchangeably. If not, and therefore the authors need to make the distinction between turbulence and wind velocity measurement with more clarity.*

The algorithm we use is that of Wetz and Wildmann (2022), which inputs the measured avionics data and outputs the converted wind speeds. We will add this information in the abstract of a revised manuscript: "To date, the coefficients for the transformation terms used in our algorithm for deriving wind speeds from avionic data, have only been determined via calibration flights in the free field."

The terminology 'wind' is used as the subordinate concept of flow velocity in the atmosphere. Turbulence is a derived parameter that is derived from the variance of the flow velocity in a range of scales that can be related to turbulent motion of the flow. In the context of the submitted manuscript, the turbulence measurement is based entirely on the wind measurement by analyzing the measured wind with regard to the turbulence it contains. Accordingly, properties of the wind measurement necessarily also affect the turbulence measurement, which is why a distinction is not required here.

3. *Line 10: The manuscript states, "our analyses show that the uncertainty depends on the wind speed magnitude and increases with higher wind speeds, resulting in an overall root-mean squared error (RMSE) of less the 0.2 m s-1." However, it is not explicitly stated which type of uncertainty the authors are referring to.*

The uncertainty is referring to the overall RMSE of all wind speeds measured in the flights with steady held wind speeds and the UAS flying in weather vane mode, which is the standard setup. We will formulate this more precisely in a revised manuscript.

4. *Line 12: The manuscript states, "The maximal RMSE occurs in the most extreme velocity steps (i.e., a lower speed of 5 m s-1 and an amplitude of 10 m s-1) and exceeds 1.3 m s-1. This result seems to contradict the result reported in Line 10.*

In our opinion, the fact that the resulting maximum RMSE is above the resulting overall RMSE is not a contradiction, but is in the nature of a maximum. However, we recognize that the upper wind speed of the most extreme velocity steps of 15 m s$^{-1}$ is not the highest wind speed that we have reached in our measurements. Accordingly, additional information that the uncertainty is also higher during extreme gusts, and not exclusively at higher wind speeds, is helpful. Consequently, we will note this in the manuscript. Nevertheless, we do not see a contradiction, as we do not state that higher wind speeds are the only determining factor. Also, the term of the overall RMSE might cause confusion here. As stated under the comment above, we will also be more precise about this.

5. *Figure 1: It would be helpful for the authors to denote the distance between the points a, b, and c, as well as the position of all 7 CTAs and the Prandtl probe in Figure 1. Additionally, since the calibration experiments were performed in a wind*

*tunnel with an open test section, were any experiments performed to quantify the wind field differences across points b and c?*

In lines 140 and 121, we describe positions b and c in relation to position a. We find this less confusing than including it in Figure 1, where angles and distances may appear distorted due to camera optics. However, we will revisit the distance information in the figure description in a revised manuscript. Furthermore, we will label the positions of the CTAs on the schematic cross in the graphic.

Measurement setups for the reference sensors, such as those we used in our experiments, are used in a comparable manner in the multitude of wind tunnel tests at the ForWind Center. Accordingly, several investigations have already been carried out on differently placed measurement points within the measurement area of the open test section. These show that the wind field differences between points b and c are small and negligible for our measurements. This is particularly the case when the grid is constantly open, as is the case with the calibration experiments referred to by the reviewer. The area below 1.5 m distance to the outlet cannot be used for the measurements as the flow is not fully developed here; position b is 2.5 m behind the outlet.Also see the supplementary material from Neuhaus et al. (2021) which shows for the example of gusts that our measurement setup is still in the usable area of the open test section. Beyond this we checked the homogeneity of the flow with an independent flow probe, in longitudinal and lateral expansion at 9 evenly distributed positions in the wind field between the wind tunnel outlet and the CTA cross. The homogeneity in the wind field was sufficient, the standard deviation of the wind velocities in the longitudinal direction is 0.09 m s$^{-1}$.

6. *Line 100: The positional drift should be reported in units of distance (i.e., m) instead of units of speed (i.e., m s-1).*

The drift is not a constant offset between the target and actual position of the UAS, but a continuous process by which the distance between the setpoint and actual position increases over time. We did manually correct the actual position during the flights. Accordingly, we consider a specification that includes the time factor to be more correct. Units of speed fulfill this condition in contrast to units of distance.

7. *Line 101: The manuscript states, "As wind speed increases, the intensity and direction of the drift change without a discernible systematic, which required constant adjustment counteracting the drift during the test flights. These adjustments were executed by the remote pilot through a manual trim." It would be useful for the reader to know if the manual trim remained constant across all test cases, and if any experiments were performed to quantify how the manual trimming affected the accuracy of wind estimates.*

Since the intensity and direction of the drift change with changing wind speed (see quoted text), and the wind speed was not constant over the full period of any measurement, it was not possible to set a constant trim for all test cases to compensate for the drift.

No dedicated experiments were carried out to specifically quantify the effect of trim on the wind measurement, but we carried out measurements in the open field using the optical flow sensor for position control while the GPS data was also logged. The drift was compensated in the same way as in the wind tunnel using trim, i. e. position hold was improved. The UAS moved horizontally a few meters in all directions, but remained above the hover position on average.

[Figure]

**Figure 1.** Time series of UAS speed in north direction measured via optical flow vs. via GPS.

90 The UAS speeds were logged for the optical flow and the GPS, in the north and east directions (Fig. 1, 2). These velocities are integrated up to the position (see Fig. 3). Note that we use the raw GPS data in this case in order to obtain the GPS data without any bias caused by the optical flow measurement.

Both sensors overestimate the deviations of the UAS position from the hover position. However, while according to GPS the UAS moves on average within a limited range of $10^1$ m, according to optical flow the UAS has moved a distance in the dimension of $10^2$ m away from the hover position. This means that the GPS measurement is considerably closer to the actual movement of the UAS while the optical flow appears to overestimate the movements of the UAS. This means that measurements under optical flow are less accurate, both when flying with and without manual trim. However, trimming improves position hold, which in turn improves the wind measurement. Accordingly, when positioning via optical flow, more trimming leads to better comparability with GPS measurements. We agree that the reader should know how our

[Figure]

**Figure 2.** Time series of UAS speed in east direction measured via optical flow vs. via GPS.

method of optical flow and manual trimming affected the accuracy of wind estimates. We therefore will address this in a revised manuscript.

8. *Line 123: The manuscript states, "Test runs with no UAS show that all CTAs measure the equivalent wind speed with sufficient accuracy: the standard deviation of the measured wind speed of the individual CTAs is less than 0.05 m/s." Is there a figure showing these results? Why not use instead the absolute error or root mean squared error to compare the performance of CTAs? Additionally, did the authors perform an analysis to determine the error between the CTAs and the Prandtl probe?*

The standard deviation is used to determine the dispersion of the data around the mean, while the RMSE is used to measure the deviation from the reference. The measurement capabilities of the individual CTAs is considered to be equally accurate, as they are all of the same type, which is why we determine the standard deviation between them.

[Figure]

**Figure 3.** UAS position relative to initial hover position derived via optical flow vs. via GPS

However, with the CTA as the reference for the UAS measurement, the RMSE is more appropriate. We agree that for the mean wind, referencing the CTAs to the Prandtl probe using RMSE is an adequate method. Accordingly, we have attached a plot of the time series (Fig. 4), and an analysis of the standard deviation and RMSE (Fig. 5) which we will also include in the manuscript's appendix.

9. *Line 135: The manuscript states, "Careful quality checks were carried out for the CTA measurement data and corrupted data was sorted out." It would be useful for the reader to know the process or criteria that was used validate the quality of CTA measurement data.*

Initially, all test cases were carried out several times. In some time series, there was a clear offset between the wind speeds measured using CTAs and Prandtl for the same test cases, which meant that the measured wind speeds did not match the preset speeds. The measurements were therefore not reproducible. However, the UAS measurements were

[Figure]

**Figure 4.** Time series of the logarithmically increased wind speeds measured with the individual CTAs and the Prandtl probe.

reproducible and show plausible velocities in relation to the preset wind speeds. The suspected source of error in these cases is related to the wiring and thus the grounding of the sensor box of the CTAs and Prandtl probe. In short, entire time series were disregarded when the reference sensors clearly output incorrect measurement data. Since this is a common and hardly noteworthy procedure for experimental work, we will delete the passage in a revised manuscript to avoid confusion. Only the measurements for 10/11/2022 morning were excluded.

10. *Line 148: It would be useful for the reader to know which specific optimization algorithm was used to estimate the calibration coefficients for Eq. 1.*

We agree, this is useful information for the reader and will be added in the revised manuscript. We use the trust region reflective algorithm of the SciPy python library (https://docs.scipy.org/doc/scipy/reference/generated/scipy.optimize.curve_fit.html, last access: 27 May 2024).

[Figure]

**Figure 5.** Standard deviation of the CTAs $\sigma$ and RMSE of the CTAs to the Prandtl probe $\epsilon$ for the measurements without UAS, which were performed twice each day.

11. *Line 150: A reference is needed for ISO 17713-1:2007*

    We agree, although ISO 17713-1:2007 was listed in the bibliography, it was not linked in the body text. This will be added in a revised manuscript

12. *Table A1 is missing entries in column two Table A2 is missing entries in column one Table A4 is missing entries in column one*

    The first reviewer has already noted that "The way Table A1 is laid out is hard to understand especially with the empty cells". This also seems to apply to Tables A2 and A4. We will improve this for the revised manuscript.

**References**

Neuhaus, L., Berger, F., Peinke, J., Hölling M.: Exploring the capabilities of active grids, Experiments in Fluids, 62, https://doi.org/10.1007/s00348-021-03224-5, 2021.

Wetz, T. and Wildmann, N.: Spatially distributed and simultaneous wind measurements with a fleet of small quadrotor UAS, Journal of Physics: Conference Series, 2265, 022 086, https://doi.org/10.1088/1742-6596/2265/2/022086, 2022.

Wetz, T., Zink, J., Bange, J., and Wildmann, N.: Analyses of Spatial Correlation and Coherence in ABL Flow with a Fleet of UAS, Bound-aryLayer Meteorology, 2023. https://doi.org/10.1007/s10546-023-00791-4

140

---

## Referee Report (RR1)

**General comments:**

The manuscript describes, more clearly this time round, an approach to measuring wind speeds using a quadcopter UAS within a wind tunnel, emphasizing calibration and verification methods. The authors are looking to refine the calibration process for the wind measurement algorithm of their SWUF-3D UAS fleet within a controlled laboratory setting. This process is important for obtaining accurate in situ measurements of the atmosphere without waiting for favorable weather conditions for a proper calibration in the open field.

Other researchers on the topic have been discussing this method to be a simple and obvious solution, and the authors of this paper are now showing more solid method description after the first round of reviews with clear data visualizations. The authors addressed my comments and successfully resolved my concerns, making the study clearer and easier to understand after revisions. Therefore, I find the content of the paper ready for acceptance and publication after addressing the suggestions below.

**Suggestions for revision:**

Although the paper is well structured, the authors still need to make better connections between figures and the tables with the general text. Particularly, I find the captions of some figures and tables lacking information which can make the reader lose interest quickly. A typical reader will glance at the figures and tables first before reading through the entire paper, therefore it is very important to already show relevant information on the captions like symbols used and their definitions, reference to equations if any, and a description of what the presented data was used for, and some short conclusions. It'd be best to see at least 2-3 lines of captions.

---

## Author Response (AR2)

**High resolution wind speed measurements with quadcopter UAS: calibration and verification in a wind tunnel with active grid**

Johannes Kistner[1], Lars Neuhaus[2,3], and Norman Wildmann[1]

[1]Deutsches Zentrum für Luft- und Raumfahrt, Institut für Physik der Atmosphäre, Oberpfaffenhofen, Germany
[2]Carl von Ossietzky Universität Oldenburg, School of Mathematics and Science, Institute of Physics
[3]ForWind - Center for Wind Energy Research, Küpkersweg 70, 26129 Oldenburg, Germany

**Correspondence:** Johannes Kistner (johannes.kistner@dlr.de)

**1 Review Comment 1**

**1.1 Suggestions for revision**

1. *Although the paper is well structured, the authors still need to make better connections between figures and the tables with the general text. Particularly, I find the captions of some figures and tables lacking information which can make the reader lose interest quickly. A typical reader will glance at the figures and tables first before reading through the entire paper, therefore it is very important to already show relevant information on the captions like symbols used and their definitions, reference to equations if any, and a description of what the presented data was used for, and some short conclusions. It'd be best to see at least 2-3 lines of captions.*

5

   We thank the reviewer for the advice and the opportunity to improve our descriptions. We agree that some tables and figures are not sufficiently described. Accordingly, we add relevant information such as explanations of the symbols and their definitions, including references to equations for a revised manuscript. However, we refrain from using short conclusions, as these require significant shortening of the content in order to fit into the size of a caption, which then may lead to oversimplification of the matter.

10

**2 Relevant changes to the manuscript**

We list here the relevant changes to the manuscript:

1. Methods

   – Caption modifications in response to referee comments.

2. Results

   – Caption modifications in response to referee comments.

3. Appendix

   – Caption modifications in response to referee comments.